# Researching COVID to enhance recovery (RECOVER) pregnancy study: Rationale, objectives and design

**Torri D. Metz**[1]*, Rebecca G. Clifton[2], Richard Gallagher[3], Rachel S. Gross[4], Leora I. Horwitz[5], Vanessa L. Jacoby[6], Susanne P. Martin-Herz[7], Myriam Peralta-Carcelen[8], Harrison T. Reeder[9], Carmen J. Beamon[10], James Chan[9], A. Ann Chang[11], Maged M. Costantine[12], Megan L. Fitzgerald[5], Andrea S. Foulkes[9], Kelly S. Gibson[13], Nick Güthe[5], Mounira Habli[14], David N. Hackney[15], Matthew K. Hoffman[16], M. Camille Hoffman[17], Brenna L. Hughes[18], Stuart D. Katz[19], Victoria Laleau[20], Gail Mallett[21], Hector Mendez-Figueroa[22], Vanessa Monzon[6], Anna Palatnik[23], Kristy T. S. Palomares[24], Samuel Parry[25], Christian M. Pettker[26], Beth A. Plunkett[27], Athena Poppas[28], Uma M. Reddy[29], Dwight J. Rouse[30], George R. Saade[31], Grecio J. Sandoval[32], Shannon M. Schlater[33], Frank C. Sciurba[34], Hyagriv N. Simhan[35], Daniel W. Skupski[36], Amber Sowles[1], Tanayott Thaweethai[9], Gelise L. Thomas[37], John M. Thorp, Jr.[38], Alan T. Tita[39], Steven J. Weiner[2], Samantha Weigand[40], Lynn M. Yee[21], Valerie J. Flaherman[20], on behalf of the RECOVER Initiative[¶]

**Data Availability Statement:** No datasets were generated or analyzed during the current study. All relevant data from this study will be made available upon study completion.

1 Department of Obstetrics and Gynecology, University of Utah Health Hospitals and Clinics, Salt Lake City, UT, United States of America, 2 Biostatistics Center, The George Washington University, Washington, DC, United States of America, 3 Department of Child and Adolescent Psychiatry, NYU Grossman School of Medicine, New York, NY, United States of America, 4 Department of Pediatrics, New York University Grossman School of Medicine, New York, NY, United States of America, 5 Department of Population Health, New York University Grossman School of Medicine, New York, NY, United States of America, 6 Department of Obstetrics, Gynecology, and Reproductive Sciences, University of California, San Francisco, San Francisco, CA, United States of America, 7 Department of Pediatrics, Division of Developmental Medicine, University of California, San Francisco, San Francisco, CA, United States of America, 8 Department of Pediatrics, University of Alabama at Birmingham, Birmingham, AL, United States of America, 9 Department of Biostatistics, Massachusetts General Hospital, Boston, MA, United States of America, 10 Department of Maternal Fetal Medicine, WakeMed Health and Hospitals, Raleigh, NC, United States of America, 11 Women's Health Research Clinical Center, University of California, San Francisco, San Francisco, CA, United States of America, 12 Department of Obstetrics and Gynecology, The Ohio State University Wexner Medical Center, Columbus, OH, United States of America, 13 Department of Obstetrics and Gynecology, The MetroHealth System, Cleveland, OH, United States of America, 14 Division Maternal Fetal Medicine, Trihealth Good Samaritan Hospital Maternal Fetal Medicine, Cincinnati, OH, United States of America, 15 Department of Obstetrics and Gynecology, University Hospitals Cleveland Medical Center: UH Cleveland Medical Center, Cleveland, OH, United States of America, 16 Department of Obstetrics and Gynecology, Christiana Care Health System, Newark, DE, United States of America, 17 Department of Obstetrics & Gynecology, University of Colorado School of Medicine, Aurora, CO, United States of America, 18 Department of Obstetrics and Gynecology, Duke University, Durham, NC, United States of America, 19 Department of Medicine, New York University School of Medicine, New York City, NY, United States of America, 20 Department of Pediatrics, University of California, San Francisco, San Francisco, CA, United States of America, 21 Department of Obstetrics and Gynecology, Northwestern University Feinberg School of Medicine, Chicago, IL, United States of America, 22 Department of Obstetrics, Gynecology and Reproductive Sciences, University of Texas McGovern Medical School: The University of Texas Health Science Center at Houston John P. and Katherine G. McGovern Medical School, Houston, TX, United States of America, 23 Department of Obstetrics and Gynecology, Medical College of Wisconsin, Milwaukee, WI, United States of America, 24 Department of Obstetrics and Gynecology, Division of Maternal Fetal Medicine, Saint Peter's University Hospital, New Brunswick, NJ, United States of America, 25 Department of Obstetrics and Gynecology, University of Pennsylvania, Philadelphia, PA, United States of America, 26 Department of Obstetrics, Gynecology and Reproductive Sciences, Yale University School of Medicine, New Haven, CT, United States of America, 27 Department of Obstetrics and Gynecology, NorthShore University HealthSystem, Evanston, IL, United States of America, 28 Division of Cardiology, Brown University Warren Alpert Medical School, Providence, RI, United States of America, 29 Department of Obstetrics and Gynecology, Columbia University, New York City, NY, United States of America, 30 Department of Obstetrics

**Funding:** National Institutes of Health (NIH) Agreement OTA OT2HL161841 (ASF), OT2HL161847 (SDK), OT2HL156812 (N/A). https://www.nih.gov/ The funders did not and will not have a role in study design, data collection and analysis, decision to publish, or preparation of the manuscript.

**Competing interests:** I have read the journal's policy and the authors of this manuscript have the following competing interests: Dr. Metz reports personal fees from Pfizer for her role as a medical consultant for a SARS-CoV-2 vaccination in pregnancy study, grants from Pfizer for role as a site PI for SARS-CoV-2 vaccination in pregnancy study, grants from Pfizer for role as a site PI for RSV vaccination in pregnancy study, and grants from Gestvision for role as a site PI for a preeclampsia study outside the submitted work. Dr. Horwitz reported serving as a member of the National Academy of Medicine Committee on the Long-Term Health Effects Stemming from COVID-19 and Implications for the Social Security Administration. Dr. Costantine reported receiving grant support for work not related to this paper from Baxter International and Siemens Healthcare and personal consulting fees not related to this paper from Progenity and Siemens Healthcare. These disclosures do not alter our adherence to PLOS ONE policies on sharing data and materials.

and Gynecology, Brown University, Providence, RI, United States of America, **31** Department of Obstetrics and Gynecology, The University of Texas Medical Branch at Galveston, Galveston, TX, United States of America, **32** Biostatistics Center, The George Washington University, Rockville, MD, United States of America, **33** Huntsman Cancer Institute, University of Utah Health, Salt Lake City, UT, United States of America, **34** Department of Medicine, Division of Pulmonary Allergy and Critical Care Medicine, University of Pittsburgh, Pittsburgh, PA, United States of America, **35** Department of Obstetrics, Gynecology and Reproductive Sciences, University of Pittsburgh School of Medicine, Pittsburgh, PA, United States of America, **36** Department of Obstetrics and Gynecology, Weill Cornell Medicine, New York, NY, United States of America, **37** Clinical and Translational Science Collaborative of Cleveland, Case Western Reserve University, Cleveland, OH, United States of America, **38** Department of Obstetrics and Gynecology, UNC: The University of North Carolina at Chapel Hill, Chapel Hill, NC, United States of America, **39** Department of Obstetrics and Gynecology, Center for Women's Reproductive Health, University of Alabama at Birmingham, Birmingham, AL, United States of America, **40** Department of Obstetrics and Gynecology, Wright State University Boonshoft School of Medicine, Dayton, OH, United States of America

¶ Membership of the RECOVER Initiative is provided in the Acknowledgments.
* torri.metz@hsc.utah.edu

# Abstract

## Importance

Pregnancy induces unique physiologic changes to the immune response and hormonal changes leading to plausible differences in the risk of developing post-acute sequelae of SARS-CoV-2 (PASC), or Long COVID. Exposure to SARS-CoV-2 during pregnancy may also have long-term ramifications for exposed offspring, and it is critical to evaluate the health outcomes of exposed children. The National Institutes of Health (NIH) Researching COVID to Enhance Recovery (RECOVER) Multi-site Observational Study of PASC aims to evaluate the long-term sequelae of SARS-CoV-2 infection in various populations. RECOVER-Pregnancy was designed specifically to address long-term outcomes in maternal-child dyads.

## Methods

RECOVER-Pregnancy cohort is a combined prospective and retrospective cohort that proposes to enroll 2,300 individuals with a pregnancy during the COVID-19 pandemic and their offspring exposed and unexposed in utero, including single and multiple gestations. Enrollment will occur both in person at 27 sites through the *Eunice Kennedy Shriver* National Institutes of Health Maternal-Fetal Medicine Units Network and remotely through national recruitment by the study team at the University of California San Francisco (UCSF). Adults with and without SARS-CoV-2 infection during pregnancy are eligible for enrollment in the pregnancy cohort and will follow the protocol for RECOVER-Adult including validated screening tools, laboratory analyses and symptom questionnaires followed by more in-depth phenotyping of PASC on a subset of the overall cohort. Offspring exposed and unexposed in utero to SARS-CoV-2 maternal infection will undergo screening tests for neurodevelopment and other health outcomes at 12, 18, 24, 36 and 48 months of age. Blood specimens will be collected at 24 months of age for SARS-CoV-2 antibody testing, storage and anticipated later analyses proposed by RECOVER and other investigators.

## Discussion

RECOVER-Pregnancy will address whether having SARS-CoV-2 during pregnancy modifies the risk factors, prevalence, and phenotype of PASC. The pregnancy cohort will also

establish whether there are increased risks of adverse long-term outcomes among children exposed in utero.

### Clinical Trials.gov Identifier

Clinical Trial Registration: http://www.clinicaltrials.gov. Unique identifier: NCT05172011.

## Introduction

The disease caused by Severe Acute Respiratory Syndrome Coronavirus-2 (SARS-CoV-2), coronavirus disease 2019 (COVID-19), results in both short-term and long-term sequelae, and many people with COVID-19 experience persistent health effects for months beyond initial infection [1–4]. Thus, evidence regarding the post-acute sequelae of SARS-CoV-2 (PASC) or "long COVID" is urgently needed to guide clinical care and inform public health policy.

Pregnant individuals may experience different long-term effects from COVID-19 than non-pregnant individuals due to the immunomodulatory effects of pregnancy. In a publication from the PRIORITY study consisting of pregnant individuals infected in 2020, 25% with COVID-19 experienced prolonged viral symptoms 8 weeks after infection, with median time to resolution of initial infection of 37 days [5]. Data beyond those first 8 weeks are not available. Thus, it remains unknown if there is a differential likelihood of PASC among individuals who contract SARS-CoV-2 during pregnancy compared with non-pregnant reproductive age females.

In addition, the offspring of patients with COVID-19 during pregnancy may be at increased risk of both short and long-term morbidity related to in utero exposure to SARS-CoV-2, resulting from inflammation and vascular changes in the placenta [6], as evidenced by investigation of the effects of other viruses and in the setting of placental insufficiency. The effects of SARS-CoV-2 on the health of pregnant individuals and their offspring may vary based on SARS-CoV-2 variant, gestational age at infection, and maternal health and vaccination status [7–12]. Existing data demonstrate that the Delta variant results both in the highest risk of maternal critical illness and stillbirth, and in the most profound placental effects [7, 8]. Vaccination has been found to be protective against both severe maternal COVID-19 and adverse pregnancy outcomes, and may also influence the incidence of PASC in this population [9–11].

The National Institute of Health's (NIH) RECOVER initiative has a unique opportunity to study the post-acute sequelae of SARS-CoV-2 acquired in pregnancy. RECOVER-Pregnancy will enroll maternal-child dyads to investigate the effects of SARS-CoV-2 acquired during pregnancy on the long-term health of both the pregnant person and their offspring who were exposed to SARS-CoV-2 in utero. We hypothesize that the incidence of persistent long-term sequelae among pregnant people will vary from that of non-pregnant reproductive age females given the unique changes in physiology, hormone levels, and inflammatory milieu in pregnancy. In addition, we hypothesize that the offspring of individuals with SARS-CoV-2 infection in pregnancy will have worse developmental outcomes when compared with unexposed offspring delivered over the same time period. The overarching goal of RECOVER-Pregnancy is to describe PASC in people who are infected with SARS-CoV-2 in pregnancy and examine the outcomes of their offspring compared with children who were not exposed to SARS-CoV-2 in utero.

## Materials and methods

### Objectives

RECOVER-Pregnancy cohort will enroll maternal-child dyads exposed and unexposed to SARS-CoV-2 during pregnancy, with pregnant individuals participating in RECOVER-Adult and offspring participating as the in utero exposed subgroup of RECOVER-Pediatric.

The broad objectives of both the RECOVER-Adult and RECOVER-Pediatric studies are to: (1) characterize the incidence and prevalence of sequelae of SARS-CoV-2 infection; (2) characterize the spectrum of clinical symptoms, subclinical organ dysfunction, natural history, and distinct phenotypes identified as sequelae of SARS-CoV-2 infection; (3) identify demographic, social, and clinical risk factors for PASC and PASC recovery, and (4) define the biological mechanisms underlying pathogenesis of PASC.

The study protocols for both RECOVER-Adult and RECOVER-Pediatric cohorts are published separately [13, 14]. The RECOVER-Pregnancy cohort is summarized here. The primary aims of the adult portion of the RECOVER-Pregnancy cohort include: (1) describing PASC onset, symptoms and severity by comparing pregnant individuals with SARS-CoV-2 with pregnant individuals without SARS-CoV-2, (2) characterizing risk factors for PASC onset, symptoms and severity following SARS-CoV-2 infection during pregnancy by comparing infected pregnant individuals with and without particular risk factors; and (3) evaluating if contracting SARS-CoV-2 in pregnancy is associated with a differential risk of developing PASC compared with non-pregnant reproductive age females. The primary aims of the pediatric portion of the RECOVER-Pregnancy cohort include characterizing the clinical manifestations of exposure to SARS-CoV-2 infection during pregnancy on child physical health and behavioral and developmental outcomes, by comparing children with and without maternal history of SARS-CoV-2 infection during pregnancy.

## Study design and setting

The NIH RECOVER study is a retrospective and prospective longitudinal meta-cohort study design in which individuals with and without exposure to SARS-CoV-2 infection are enrolled and followed over a maximum of 4 years. For RECOVER-Pregnancy, individuals with and without SARS-CoV-2 infection during pregnancy will be included. Individuals with a SARS-CoV-2 infection during pregnancy will be further classified as acute (infection within the past 30 days at the time of enrollment) and post-acute (more than 30 days since infection). The RECOVER-Pregnancy cohort follows both the adult protocol and the part of the pediatric protocol that is specific to in utero exposure to SARS-CoV-2.

Mother-child dyads for the RECOVER-Pregnancy cohort are recruited in person at 12 *Eunice Kennedy Shriver* National Institute of Child Health and Human Development (NICHD) Maternal-Fetal Medicine Units (MFMU) Network Centers which include 26 individual hospital sites (encompassing community and academic hospitals) and remotely nationwide by the University of California San Francisco (UCSF). A list of study sites can be obtained at https://recovercovid.org or in S1 Checklist. For participants recruited within the 26 individual hospital sites of the MFMU Network, recruitment occurs among individuals who were enrolled in the NICHD MFMU Gestational Research Assessments for COVID-19 (GRAVID) study [15, 16], a previously established cohort of individuals with and without SARS-CoV-2 during pregnancy over the COVID-19 pandemic, and new prospective enrollments of individuals with and without SARS-CoV-2 in pregnancy to encompass all variants of SARS-CoV-2. Study procedures for participants recruited by NICHD MFMU Network sites will be completed through a combination of remote survey assessments and in-person study visits. For those recruited remotely by the study team at the UCSF, recruitment occurs among individuals who were enrolled in the PRIORITY study, a previously established nationwide cohort of individuals with SARS-CoV-2 during pregnancy, as well as through new nationwide outreach including social media outreach, referrals from the NIH RECOVER website, screening of the UCSF electronic health record and from RECOVER-Adult cohort sites that are not also participating in the pregnancy cohort. Participants recruited by UCSF will have study

procedures such as anthropometric measurements and specimen collections completed during home visits while surveys are completed remotely. Individuals who are able to travel to UCSF may have study procedures completed in person. Participants can only be enrolled once to avoid overlap in recruitment across the RECOVER-Pregnancy sites.

## Study population

Anyone with SARS-CoV-2 infection during pregnancy (defined as meeting World Health Organization criteria [17] for suspected, probable or confirmed SARS-CoV-2 infection), or with documented lack of SARS-CoV-2 exposure during pregnancy, irrespective of pregnancy outcome (loss, termination, or live birth) is eligible to enroll in RECOVER-Pregnancy (See RECOVER-Adult protocol for detailed definition) [13]. The index date for infection is set as the date of the first infection as determined by the date of a positive SARS-CoV-2 test, or set at 90 days prior to antibody test date for those with positive antibodies who had an asymptomatic prior infection. Individuals with SARS-CoV-2 in pregnancies resulting in a live birth are only eligible for enrollment into one of the RECOVER pregnancy cohorts to facilitate co-enrollment of the offspring in the in utero exposure part of the pediatric protocol. Individuals with pregnancies resulting in pregnancy loss or termination are eligible to enroll at any adult cohort site. Pregnancies and pregnancy outcomes for all adults in the RECOVER-Adult cohort over the longitudinal follow-up period for RECOVER will also be recorded.

Those who had a pregnancy during the study enrollment window without SARS-CoV-2 infection are eligible for enrollment as uninfected participants. Uninfected participants must have no reported history of SARS-CoV-2 infection, and be PCR negative and SARS-CoV-2 antibody negative at the time of enrollment. For the pregnancy cohort, index date for the post-acute uninfected participants is set at the date of delivery as the uninfected period of interest is the pregnancy (e.g. uninfected participants should not have been infected during pregnancy). The recruitment strategy seeks to have infected and uninfected individuals enrolled within the same time frame to reduce potential effects of societal modifications during the pandemic which may also affect neurodevelopment (e.g., social isolation, disruptions to education and childcare services, financial parental stressors, etc) [18, 19].

## Sample size

Overall 2,300 adults with or without SARS-CoV-2 during pregnancy will be enrolled in RECOVER-Pregnancy along with their offspring. Of the 2300 pregnant individuals enrolled in the adult cohort, we plan to enroll 1867 infected with SARS-CoV-2 during pregnancy and 433 uninfected during pregnancy. Because pregnant participants are enrolling as dyads with their offspring, we expect to enroll corresponding numbers of offspring exposed to SARS-CoV-2 during pregnancy and unexposed during pregnancy, allowing for variability due to factors such as twins or multiple gestations and pregnancy loss. Of the adults enrolling in RECOVER-Pregnancy with SARS-CoV-2 infections, we plan for 15% of them to be enrolled during a period of acute infection (within 30 days of positive SARS-CoV-2 test).

For each comparison underlying the primary study objectives, we computed minimum effect sizes detectable with 90% power and a type-1 error rate of 0.01, for the full target sample sizes, as well as for only 25% of the overall cohort, as would be expected in a subgroup analysis. These results are also reported in Table 1 in terms of odds ratios corresponding with planned logistic regression analyses, and equivalently as risk differences to further facilitate interpretation. The minimum detectable PASC risk difference between pregnant participants with and without infection is 8.4% (15.8% in a 25% subgroup), assuming the risk of developing PASC among participants with infection is 25%. Assuming the prevalence of a given risk factor

Table 1. Sample size calculations for RECOVER-Pregnancy cohort.

| Comparison groups | | Outcome of interest | Assumptions | Minimum Detectable Difference | | Minimum Detectable Odds Ratio | |
|---|---|---|---|---|---|---|---|
| | | | | (90% power, 0.01 type I error rate) | | (90% power, 0.01 type I error rate) | |
| | | | | Full Sample | 25% Subgroup | Full Sample | 25% Subgroup |
| Pregnant infected (n = 1867) | Pregnant uninfected (n = 433) | Risk of PASC | Risk of PASC in infected: 25% | 8.4% | 15.8% | 1.68 | 3.28 |
| Pregnant infected w/ RF[a] (n = 373) | Pregnant infected w/out RF[a] (n = 1494) | Risk of PASC | Prevalence of RF[a]: 20% | 9.5% | 17.4% | 1.67 | 2.98 |
| | | | Risk of PASC in infected, RF[a]+: 30% | | | | |
| Pregnant infected (n = 1867) | Non-pregnant infected reproductive age female (n = 3283) | Risk of PASC | Risk of PASC in non-pregnant infected: 25% | 4.7% | 8.9% | 1.31 | 1.74 |
| | | | 25% of non-pregnant infected are reproductive age females | | | | |
| Infant exposed in-utero (n = 1867) | Infant not exposed in-utero (n = 433) | Continuous biomarker or development score | Equal variances | 0.206 SDs[a] | 0.413 SDs[a] | - | - |

[a] (RF: risk factor; RF+: risk factor positive; SD: standard deviation)

among pregnant participants with infection is 20%, and the risk of PASC in pregnant participants with infection who have the risk factor is 30%, the minimum detectable PASC risk difference between infected participants with and without the risk factor is 9.5% (17.4% in a 25% subgroup). Assuming 25% of non-pregnant adult participants are female and of reproductive age, the minimum detectable PASC risk difference between SARS-CoV-2 infected reproductive age female participants with and without active pregnancy at infection is 4.7% (8.9% in a 25% subgroup). Finally, the minimum detectable difference in a continuous biomarker or development score between offspring exposed to SARS-CoV-2 in utero and those unexposed during pregnancy is 0.206 standard deviations (0.413 in a 25% subgroup).

Importantly, adult RECOVER-Pregnancy cohort participants will be included in the planned primary analyses for RECOVER-Adult. Classification of adult pregnancy cohort participants will vary depending on the analysis being performed. For example, an adult participant who was uninfected during pregnancy will remain in the uninfected comparison group for all analyses examining outcomes of the offspring, as the offspring will not be exposed in utero. However, some of the enrolled adult pregnancy participants who were enrolled as uninfected participants will acquire SARS-CoV-2 infection during the 4-year longitudinal follow-up period. These participants will cross over to the infected group for the adult cohort, and will be analyzed as infected in analyses assessing only the adult outcomes from that time period forward, but their offspring will remain in the unexposed pediatric cohort. Thus, the RECOVER Data Resource Center (DRC) will classify pregnant individuals with SARS-CoV-2 and their offspring intentionally and separately for each analysis depending on the outcome being examined.

## Study procedures

Details of protocol development are included separately in study protocol manuscripts for the RECOVER-Adult and RECOVER-Pediatric protocols [13, 14]. Development of the pregnancy cohort study procedures and protocol occurred concurrently with the principal investigators

| eCRF | Baseline | 3m | 6m | 9m | 12m | 15m | 18m | 21m | 24m | 27m | 30m | 33m | 36m | 39m | 42m | 45m | 48m |
|---|---|---|---|---|---|---|---|---|---|---|---|---|---|---|---|---|---|
| | | | | | | | | | Time Point after index date | | | | | | | | |
| Enrollment | ● | | | | | | | | | | | | | | | | |
| Tier 1-2 Consent | ● | | | | | | | | | | | | | | | | |
| Identity | ● | | | | | | | | | | | | | | | | |
| Visit | ● | ● | ● | ● | ● | ● | ● | ● | ● | ● | ● | ● | ● | ● | ● | ● | ● |
| Comorbidities | ● | ● | ● | ● | ● | ● | ● | ● | ● | ● | ● | ● | ● | ● | ● | ● | ● |
| COVID Treatment* | ● | | | | | | | | | | | | | | | | |
| Medications | ● | | | | | | | | | | | | | | | | |
| Change in Medications | | ● | ● | ● | ● | ● | ● | ● | ● | ● | ● | ● | ● | ● | ● | ● | ● |
| Demographics | ● | | | | | | | | | | | | | | | | |
| PASC Symptoms | ● | ● | ● | ● | ● | ● | ● | ● | ● | ● | ● | ● | ● | ● | ● | ● | ● |
| Vaccination Status | ● | ● | ● | ● | ● | ● | ● | ● | ● | ● | ● | ● | ● | ● | ● | ● | ● |
| Social Determinants of Health | ● | | | | | | | | | | | | | | | | |
| Social Determinants Follow-up | | ● | ● | ● | ● | ● | ● | ● | ● | ● | ● | ● | ● | ● | ● | ● | ● |
| Alcohol/Tobacco | ● | | | | | | | | | | | | | | | | |
| Alcohol/Tobacco Follow-up | | ● | ● | ● | ● | ● | ● | ● | ● | ● | ● | ● | ● | ● | ● | ● | ● |
| Disability | ● | | | | | | | | | | | | | | | | |
| Pregnancy | ● | | | | | | | | | | | | | | | | |
| Pregnancy Follow-up | | ● | ● | ● | ● | ● | ● | ● | ● | ● | ● | ● | ● | ● | ● | ● | ● |
| Tier 1 office visit | ● | | ● | | ● | | | | ● | | | | ● | | | | ● |
| Biospecimens | ● | ● | ● | | ● | | | | ● | | | | ● | | | | ● |
| Lab Results | ● | ● | ● | | ● | | | | ● | | | | ● | | | | ● |
| Tier 2/Tier 3 Tests | | | | | | | | | | | | | | | | | |

\* COVID Treatment not collected on people without infection

**Legend**
● Completed by research coordinator
● Completed by participant
● Completed by research coordinator with review/validation by participant

**Fig 1. Study procedures for RECOVER-Adult.** A schematic of the relationship between pregnant individuals and their offspring, and the distribution of acute and post-acute SARS-CoV-2 infection is included in Fig 2. For adult participants, the pregnant cohort simply represents a predefined portion of the participants to ensure that PASC following SARS-CoV-2 infection during pregnancy can be examined and to examine differential effects when SARS-CoV-2 is acquired during pregnancy.

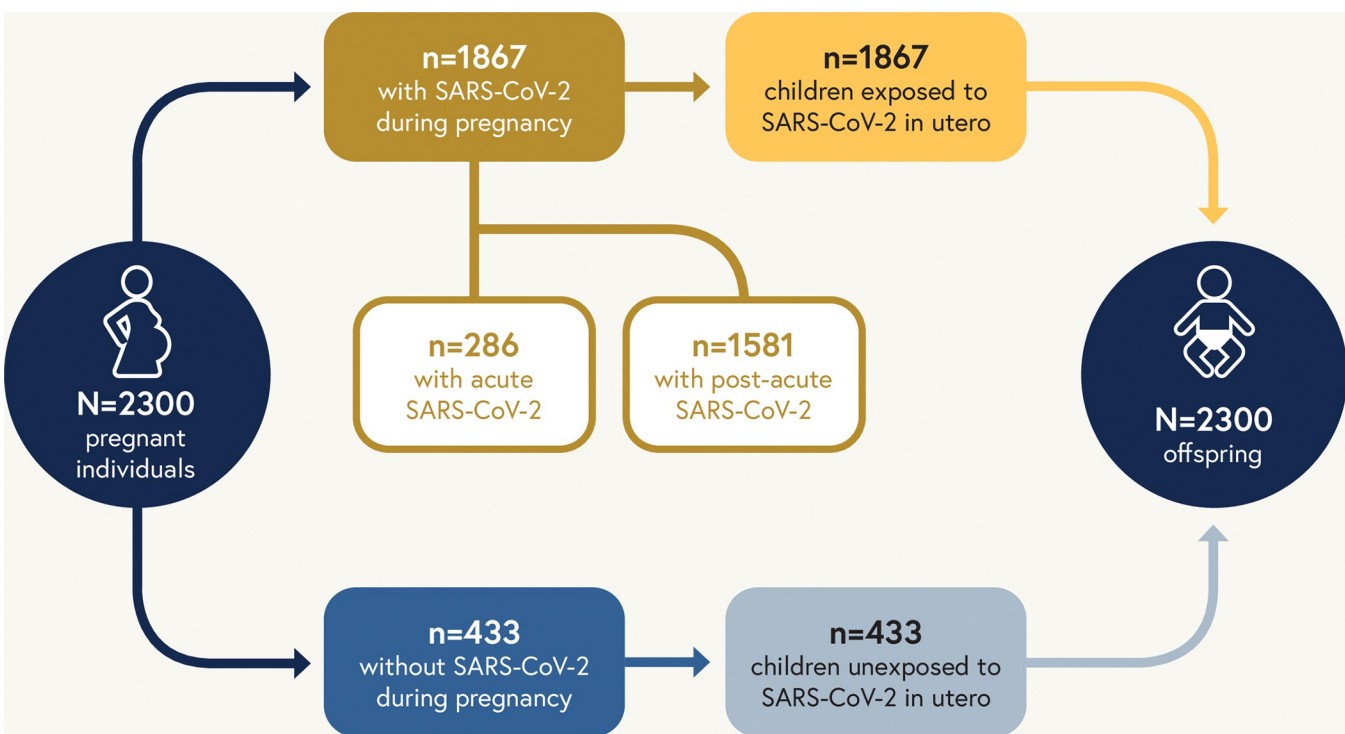

**Fig 2. Schematic representation of planned enrollment for RECOVER-Pregnancy cohort.** RECOVER-Pregnancy cohort includes pregnant individuals (both with and without SARS-CoV-2 during pregnancy) and the pediatric in utero exposure cohort. Acute infection is infection within the last 30 days and post-acute infection is infection during pregnancy, but at a time that was more than 30 days from the date of enrollment.

for the pregnancy cohort participating in the development of both the RECOVER-Adult and RECOVER-Pediatric protocols. Study procedures for RECOVER-Adult are included in Fig 1.

## Adult cohort

The primary endpoint for the adult portion of the RECOVER-Pregnancy cohort mirrors that of the larger adult cohort and is defined as the presence of incident or prevalent candidate PASC symptoms over time. Secondary endpoints include biological and recovery trajectories from SARS-CoV-2 infection, documentation of organ injury, and incident clinical diagnoses.

Tier 1 RECOVER adult study procedures consist of completion of symptom surveys for a variety of PASC symptoms, height, weight and vital signs measurements, sit to stand and active stand evaluations, a core set of laboratory studies, and biospecimen collection. Each Tier 2 test is expected to be completed by approximately 30% of adult participants. Tier 2 testing can be triggered by abnormal values or responses in Tier 1; in addition, tier 2 testing is triggered in a randomly selected group of uninfected participants for comparison. Adult Tier 3 tests are both more invasive and more time consuming. Each tier 3 test is expected to be performed in approximately 20% of RECOVER participants. Participants may select to opt out of tier 2 and tier 3 tests without withdrawing from the study.

Adult participants in the pregnancy cohort complete all of the Tier 1 study procedures as defined in the adult cohort study design manuscript [13]. Similarly, pregnancy cohort participants are eligible for all of the Tier 2 and 3 study procedures with the exception of those detailed in Table 2 as these were felt to have unacceptable risk to the participant, the fetus, or

**Table 2. RECOVER-Adult study procedure modifications for pregnant, breastfeeding, or postpartum individuals.**

| Tier of Test | Adult Study Procedure Modification | Excluded Population | Justification for Modification |
|---|---|---|---|
| Tier 2 | Volumetric non-contrast chest CT | Pregnant and <3 months postpartum | Unacceptable risk of radiation exposure to the fetus without direct benefit |
| Tier 2 | Dual energy chest CT with contrast | Pregnant and <3 months postpartum | Unacceptable risk of radiation exposure to the fetus without direct benefit |
| Tier 3 | MRI brain with gadolinium | Pregnant and <3 months postpartum | Unacceptable risk of gadolinium exposure for fetus and neonate (via breastmilk) with no direct participant benefit |
| Tier 3 | Cardiac imaging with meta-iodobenzylguanidine (mIBG) | Pregnant and <3 months postpartum | Unacceptable risk of mIBG exposure for fetus and neonate (via breastmilk) with no direct participant benefit |
| Tier 3 | Gastric emptying study | Pregnant and <3 months postpartum | Physiologic changes in pregnancy and early postpartum period with gut motility precludes accurate interpretation |
| Tier 3 | Skin biopsy | Pregnant and <3 months postpartum | Invasive procedure without direct participant benefit |
| Tier 3 | Muscle biopsy | Pregnant and <3 months postpartum | Invasive procedure without direct participant benefit |
| Tier 3 | Lumbar puncture | Pregnant and <3 months postpartum | Invasive procedure without direct participant benefit |
| Tier 3 | Tilt table testing | Pregnant and <3 months postpartum | Physiologic changes in pregnancy and early postpartum period with vasodilatory response precludes accurate interpretation |
| Tier 3 | Full cardiopulmonary exercise testing | Pregnant and postpartum | Exercise tolerance differs during pregnancy and early postpartum period which precludes accurate interpretation |
| Tier 3 | Bronchoscopy | Pregnant and postpartum | Invasive procedure and anesthesia without direct participant benefit |
| Tier 3 | Right heart catheterization | Pregnant and postpartum | Invasive procedure and anesthesia without direct participant benefit |
| Tier 3 | Upper endoscopy | Pregnant and postpartum | Invasive procedure and anesthesia without direct participant benefit |
| Tier 3 | Colonoscopy with or without biopsy | Pregnant and postpartum | Invasive procedure and anesthesia without direct participant benefit |
| Tier 3 | Cardiac imaging with meta-iodobenzylguanidine (mIBG) | Breastfeeding | Unacceptable risk of contrast exposure to neonate through breastmilk with no direct participant benefit |
| Tier 3 | Gastric emptying study | Breastfeeding | Unacceptable risk of contrast exposure to neonate through breastmilk with no direct participant benefit |

neonate for this observational study. Similarly, triggers for some of the Tier 2 and Tier 3 tests have been modified for those who are pregnant, postpartum (within 3 months of delivery date), or breastfeeding as they would not be appropriate triggers during these time periods (Table 2). For example, weight loss is not used as a test trigger for participants who are postpartum as weight loss is anticipated postpartum.

Pregnancy characteristics and pregnancy outcome data will be ascertained by maternal self-report through survey questions following the end of pregnancy. The survey questions will be validated through manual medical record abstraction at the NICHD MFMU Centers as medical record data for participants are readily available at the Centers. Pregnancy characteristics and outcome data that will be collected are included in Box 1. Additional important covariates that are being collected include trimester of pregnancy at time of infection, severity of infection and treatment for infection.

---

### Box 1. Pregnancy outcome data ascertained for RECOVER-Pregnancy cohort

Estimated date of delivery

Actual delivery date

Number of prior pregnancies and outcome of prior pregnancies

Number of fetuses in index pregnancy (either infected or uninfected with SARS-CoV-2)

Live birth, stillbirth, miscarriage, or termination

Birthweight of fetus(es)

Gestational diabetes

Hypertensive disorder of pregnancy

Cesarean birth

Diagnosis of intraamniotic infection

Placental abruption

Administration of magnesium sulfate

Administration of antenatal corticosteroids

Postpartum hemorrhage

ICU admission during delivery hospitalization

---

### Pediatric in utero exposure cohort

All children will participate in Tier 1 assessments, which include broad health screening measures collected remotely via electronic surveys or telephone interviews, and Tier 2 assessments, which include more in-depth clinical assessments conducted either at in-person visits or via videoconferencing. Tier 1 assessments occur at 12, 18, 24, 36 and 48 months of age, and Tier 2 assessments occur at 24, 36 and 48 months of age (Fig 3).

Both Tiers 1 and 2 focus primarily on child developmental outcomes. Tier 1 includes developmental screening across the domains of communication, gross motor, fine motor, problem

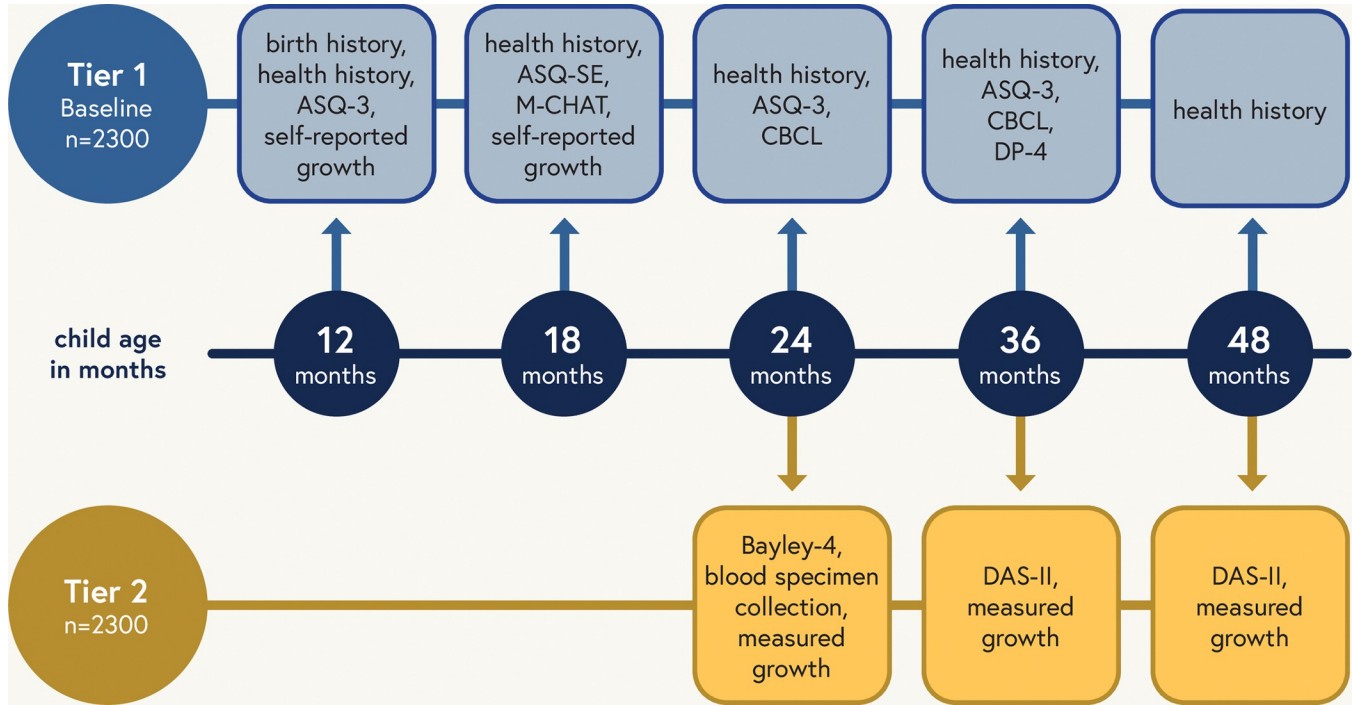

**Fig 3. Study assessment schedule for children enrolled in RECOVER-Pregnancy.** Visual depiction of the study assessment scheduled for the children enrolled in RECOVER-Pregnancy or the in utero exposure cohort of RECOVER.

solving, and personal-social development using the Ages and Stages Questionnaire—3rd Edition (ASQ-3) [20–22], social-emotional screening using the Ages and Stages Questionnaires: Social-Emotional, 2nd Edition (ASQ:SE-2) and standardized screening for Autism Spectrum Disorder (ASD) using the Modified Checklist for Autism in Toddlers Revised with Follow up (MCHAT-RF) [23] (Fig 3). Child socioemotional and behavioral status will be assessed using the Child Behavior Checklist (CBCL) [24]. In addition, adaptive behavior, communication, cognitive development, social emotional development, and physical development, will be assessed using the Developmental Profile-4 (DP-4) [25] by direct caregiver interview (Fig 3).

Tier 2 assessments will include an evaluation of child development using the Bayley Scales of Infant and Toddler Development, 4th Edition (Bayley-4) [26] and the Differential Abilities Scales, 2nd Edition (DAS-II) [27]. The Bayley-4 is a comprehensive developmental assessment tool which will evaluate cognitive, language, and motor development for RECOVER. The DAS-II is a comprehensive assessment that provides insight into how a child processes information and problem-solving, related to verbal comprehension and nonverbal and spatial reasoning. Training for administration of the Bayley-4, DAS-II and DP-4 will be conducted centrally by the RECOVER Clinical Science Core (CSC) Neurocognitive Leadership Team, and videos of mock administrations will be reviewed for all examiners to ensure validity of the administered exam prior to performing any exams for the purposes of the RECOVER study.

In addition to the neurocognitive outcomes, broader child health outcomes are assessed as well. Tier 1 surveys assess sociodemographic information, medical history and special health care needs using items from the National Surveys of Children's Health [28], SARS-CoV-2 infection history, related conditions (e.g., Multisystem inflammatory syndrome in children (MIS-C), Postural Orthostatic Tachycardia Syndrome, and Long COVID diagnoses), COVID-19 testing and vaccine history, COVID-19-related symptoms (acute and long-term), COVID-

19 health consequences (e.g., diet, physical activity, sleep, screen time, schooling, parenting) and social determinants of health (e.g., early childhood experiences or stressors). Brief infant Sleep Questionnaire (BISQ) [29] is also collected at 24 months of age.

Other Tier 2 assessments include anthropometric measurements, such as child weight, length/height, and head circumference measured using well calibrated, quality instruments and standardized techniques, to ultimately assess child weight trajectories, weight status and cardiometabolic risk. A sample of blood will be obtained at 24 months for SARS-CoV-2 antibody testing and central biobanking. Blood will be collected at 24 months by a trained and experienced pediatric phlebotomist, or through remote at-home collection using a TASSO M20 device, which collects capillary blood using 4 volumetric sponges that each hold 17.5μL of blood (70 μL total). If using the Tasso M20 device, one sponge is used for SARS-CoV-2 spike and nucleocapsid antibody testing and remaining sponges are banked for future use. If using in-person phlebotomy, EDTA and SST tubes will be collected ensuring that the total blood volume is less than 2 mL per kg of body weight.

## Data collection, management and quality assurance

All data are collected through secure entry into a single instance of REDCap created by the RECOVER DRC. Oversight of data and safety is provided by the RECOVER Observational Safety Monitoring Board (OSMB) appointed by the National Institutes of Health (NIH). The purpose of the OSMB is to assure independent review as to whether study patients are exposed to unreasonable risk because of study participation, and to monitor study progress and integrity. Data collection and quality assurance are further detailed in the RECOVER-Adult cohort study design manuscript [13].

## Statistical analysis

The point prevalence of PASC symptoms, defined as the proportion of participants reporting a symptom at a given follow-up time point among those remaining in RECOVER, will be calculated across the follow-up period for participants with infection during pregnancy and without infection during pregnancy, both overall and among those whose pregnancy has ended (termination, miscarriage, or delivery). Odds ratios (ORs) adjusted for demographic factors will be reported. A paradigm for identifying PASC cases will be developed in the broader adult cohort as described in the RECOVER-Adult study design paper, and evaluated in the population with SARS-CoV-2 infection during pregnancy along with alternative clinically-derived PASC definitions.

Logistic regression analyses will be conducted to investigate associations between cumulative incidence of PASC and clinical factors including vaccination status, demographics, and social determinants of health among participants with infection during pregnancy, as well as comparing those infected during pregnancy with infected non-pregnant reproductive age females. Sensitivity analyses assessing the influence of pregnancy factors such as gestational age at infection, and timing of follow-up visits relative to the end of pregnancy, will be considered.

Measures of child physical health and behavioral and developmental outcomes will be summarized in the in utero exposure to SARS-CoV-2 cohort overall and by maternal infection status during pregnancy. Two-sample, two-sided t-tests will be used to compare cross-sectional means of continuous outcomes between groups, and chi-squared tests will be used to compare categorical outcomes between groups. Multivariable linear and generalized linear regression models will also be fit to adjust for relevant baseline factors that are different between groups. For outcomes measured at multiple visits, trajectories will be modeled and compared using

linear and generalized linear mixed effects models. Statistical analyses will be performed using R statistical software [30].

## Safety and possible risks

All RECOVER activities are approved by a single institutional review board (IRB), which is the New York University IRB. All of the RECOVER-Pregnancy sites will establish reliance on the single IRB for RECOVER. Any protocol modifications will be approved by the single IRB and communicated with all RECOVER investigators and research staff by the CSC. Written informed consent will be obtained for all participants by research personnel at participating RECOVER-Pregnancy sites. Participants will have the opportunity to opt out of storage of specimens for future use and for genetic analyses.

For both the pediatric and adult cohort portions of the pregnancy cohort, results from laboratory testing and other tests performed as study procedures are monitored centrally by the CSC, DRC, and the individual site PIs along with their research teams. For each lab result, the research team determines if the result is normal and abnormal, and if abnormal whether it is clinically actionable. For pregnant participants, many of the laboratory results fall outside of established lab normal values; yet, are normal for pregnant individuals. Thus, thresholds for clinical action differ for this special population and the need for clinical action will be determined by the investigators at each site who are obstetricians or maternal-fetal medicine subspecialists.

If the neurodevelopmental test score or social emotional score falls in a range where additional monitoring or referral are recommended, results of screening tests completed in Tier 1 (ASQ-3, ASQ:SE-2, MCHAT R/F) or assessment tests completed in Tier 2 (Bayley-4, DAS-II and DP-4) will be provided to the caregiver of the participant with a recommendation to follow-up with the child's primary health care provider. Ranges of concern will be determined by psychologists and developmental-behavioral pediatricians, and standardized across the cohort.

## Discussion

The NIH RECOVER-Pregnancy cohort will answer critical questions regarding the incidence and trajectory of PASC following SARS-CoV-2 infection during pregnancy, and the effects of in utero exposure to SARS-CoV-2 infection on offspring development. There are little data regarding the incidence of PASC in a population who acquired SARS-CoV-2 during pregnancy. Initial data from the PRIORITY Registry suggest persistence of symptoms among pregnant individuals for weeks to months after SARS-CoV-2 infection [5]. As pregnant individuals have alterations in inflammatory response and hormone levels, and have been noted to have an increased risk of severe or critical illness with SARS-CoV-2 infection, it is plausible that the rates of PASC could be different in this group [4, 31, 32]. Differences in the prevalence or symptoms of PASC among those who acquire SARS-CoV-2 in pregnancy compared with non-pregnant reproductive age females could help with determination of the underlying pathophysiology of PASC.

Recent research suggests that being born during the pandemic is associated with lower scores on neurodevelopmental screening than in historical controls; however, exposure to SARS-CoV-2 infection alone was not associated with a difference from unexposed children living through the pandemic [18, 19]. A large administrative database study by Edlow and colleagues found an association between exposure to SARS-CoV-2 in utero and neurodevelopmental diagnoses, but was limited by the possibility of ascertainment bias, because those exposed to SARS-CoV-2 may have been more likely to be followed and referred [33]. The RECOVER-Pregnancy study will add to this available evidence by comparing

children who were SARS-CoV-2 exposed with contemporaneous unexposed children, and following them longitudinally over time. The RECOVER-Pregnancy study will also be able to examine for potential differences in subsets of the population exposed to SARS-CoV-2 in utero, such as by trimester of infection, or by maternal disease severity and treatments for SARS-CoV-2.

The design of the pregnancy cohort provides several strengths. Enrollment of maternal-child dyads across the U.S. will increase generalizability of study results, enhance racial, ethnic, and socioeconomic diversity, and enable collection of detailed information on both the mother and the child to enable robust analyses. SARS-CoV-2 is known to have disproportionately affected Black and Latino communities (especially early in the pandemic) [34, 35]; thus, the RECOVER-Pregnancy investigators are committed to reaching a diverse population for studying the long-term effects of contracting SARS-CoV-2 in pregnancy. Individuals with pregnancies across the time period of the COVID-19 pandemic will be included allowing for comparisons by major SARS-CoV-2 variants, which are known to differentially affect pregnancy outcomes [7, 8]. Allowing both in-person and remote assessments may help reach participants who could not participate in one or the other types of study visits. Developmental and socioemotional-behavioral screening and direct neurodevelopmental assessments will be performed by centrally trained and certified study personnel.

Our study design has several important limitations. First, since study procedures are extensive and require significant time, families who have less availability to attend study visits may be less likely to enroll in or more likely to be lost to follow up. This could reduce the generalizability of our results. In addition, because participants born at any time during the pandemic are eligible for enrollment, participants born earlier in the pandemic will not be able to be assessed for outcomes measured by the study at ages prior to their date of enrollment; thus, we will have smaller sample sizes available for some of the pediatric outcomes.

We anticipate that further modifications to the protocol will be made as the COVID-19 pandemic evolves and the scientific community learns more about both PASC and the effects of exposure to SARS-CoV-2 in utero on offspring. Nonetheless we have designed a rigorous study to evaluate the effect of SARS-CoV-2 on maternal-child dyads and anticipate that it will substantially further scientific understanding of the risks of contracting SARS-CoV-2 while pregnant.

## Supporting information

**S1 Checklist. SPIRIT 2013 checklist: Recommended items to address in a clinical trial protocol and related documents\*.**
(DOC)

**S1 File. RECOVER-Pregnancy cohort study sites.**
(DOCX)

**S2 File. RECOVER-Pregnancy consortium members.**
(DOCX)

## Acknowledgments

We would like to thank the National Community Engagement Group (NCEG), all patient, caregiver and community representatives, and all the participants enrolled in the RECOVER initiative.

Membership of the RECOVER Initiative Consortium is provided in S2 File.

**Disclaimer:** The content is solely the responsibility of the authors and does not necessarily represent the official views of the RECOVER program, the NIH or other funders.

## Author Contributions

**Conceptualization:** Torri D. Metz, Rebecca G. Clifton, Richard Gallagher, Rachel S. Gross, Leora I. Horwitz, Vanessa L. Jacoby, Susanne P. Martin-Herz, Myriam Peralta-Carcelen, Maged M. Costantine, Andrea S. Foulkes, Stuart D. Katz, Athena Poppas, Dwight J. Rouse, George R. Saade, Frank C. Sciurba, Tanayott Thaweethai, Alan T. Tita, Valerie J. Flaherman.

**Data curation:** Harrison T. Reeder, James Chan, Andrea S. Foulkes, Mounira Habli, Matthew K. Hoffman, M. Camille Hoffman, Stuart D. Katz, Anna Palatnik, Kristy T. S. Palomares, Dwight J. Rouse, Tanayott Thaweethai, John M. Thorp, Jr., Alan T. Tita, Valerie J. Flaherman.

**Formal analysis:** Valerie J. Flaherman.

**Funding acquisition:** Torri D. Metz, Rebecca G. Clifton, Rachel S. Gross, Leora I. Horwitz, Susanne P. Martin-Herz, A. Ann Chang, Andrea S. Foulkes, Stuart D. Katz, Dwight J. Rouse, George R. Saade, Frank C. Sciurba, Tanayott Thaweethai, Alan T. Tita, Valerie J. Flaherman.

**Investigation:** Rachel S. Gross, Leora I. Horwitz, Vanessa L. Jacoby, Susanne P. Martin-Herz, Carmen J. Beamon, Maged M. Costantine, David N. Hackney, M. Camille Hoffman, Stuart D. Katz, Gail Mallett, Hector Mendez-Figueroa, Anna Palatnik, Samuel Parry, Beth A. Plunkett, Uma M. Reddy, George R. Saade, Hyagriv N. Simhan, Amber Sowles, John M. Thorp, Jr., Alan T. Tita, Lynn M. Yee, Valerie J. Flaherman.

**Methodology:** Torri D. Metz, Richard Gallagher, Rachel S. Gross, Leora I. Horwitz, Vanessa L. Jacoby, Susanne P. Martin-Herz, Myriam Peralta-Carcelen, Harrison T. Reeder, James Chan, Andrea S. Foulkes, Stuart D. Katz, Athena Poppas, Dwight J. Rouse, Tanayott Thaweethai, Alan T. Tita, Valerie J. Flaherman.

**Project administration:** Torri D. Metz, Rebecca G. Clifton, Rachel S. Gross, Leora I. Horwitz, Harrison T. Reeder, James Chan, A. Ann Chang, Maged M. Costantine, Andrea S. Foulkes, David N. Hackney, Stuart D. Katz, Victoria Laleau, Vanessa Monzon, Anna Palatnik, Shannon M. Schlater, Frank C. Sciurba, Hyagriv N. Simhan, Amber Sowles, Tanayott Thaweethai, John M. Thorp, Jr., Alan T. Tita, Steven J. Weiner, Lynn M. Yee, Valerie J. Flaherman.

**Resources:** Rachel S. Gross, Leora I. Horwitz, Susanne P. Martin-Herz, Myriam Peralta-Carcelen, Harrison T. Reeder, James Chan, Andrea S. Foulkes, Mounira Habli, David N. Hackney, Stuart D. Katz, George R. Saade, Tanayott Thaweethai, Alan T. Tita, Samantha Weigand, Lynn M. Yee.

**Software:** Leora I. Horwitz, Harrison T. Reeder, James Chan, Andrea S. Foulkes, Tanayott Thaweethai.

**Supervision:** Torri D. Metz, Rachel S. Gross, Leora I. Horwitz, Vanessa L. Jacoby, Carmen J. Beamon, James Chan, A. Ann Chang, Maged M. Costantine, Andrea S. Foulkes, David N. Hackney, Matthew K. Hoffman, M. Camille Hoffman, Victoria Laleau, Hector Mendez-Figueroa, Vanessa Monzon, Beth A. Plunkett, Athena Poppas, Dwight J. Rouse, George R. Saade, Hyagriv N. Simhan, Daniel W. Skupski, Tanayott Thaweethai, Alan T. Tita, Samantha Weigand, Lynn M. Yee, Valerie J. Flaherman.

**Validation:** Rachel S. Gross, Leora I. Horwitz.

**Visualization:** Leora I. Horwitz, Susanne P. Martin-Herz.

**Writing – original draft:** Torri D. Metz, Rebecca G. Clifton, Richard Gallagher, Harrison T. Reeder.

**Writing – review & editing:** Torri D. Metz, Rebecca G. Clifton, Richard Gallagher, Rachel S. Gross, Leora I. Horwitz, Vanessa L. Jacoby, Susanne P. Martin-Herz, Myriam Peralta-Carcelen, Harrison T. Reeder, James Chan, Maged M. Costantine, Megan L. Fitzgerald, Andrea S. Foulkes, Kelly S. Gibson, Nick Güthe, Mounira Habli, M. Camille Hoffman, Brenna L. Hughes, Stuart D. Katz, Samuel Parry, Christian M. Pettker, Beth A. Plunkett, Uma M. Reddy, Dwight J. Rouse, George R. Saade, Grecio J. Sandoval, Hyagriv N. Simhan, Daniel W. Skupski, Tanayott Thaweethai, Gelise L. Thomas, John M. Thorp, Jr., Alan T. Tita, Steven J. Weiner, Samantha Weigand, Lynn M. Yee, Valerie J. Flaherman.

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
