## [Decision Letter · Decision Letter 0]

21 Jul 2023

PONE-D-23-11234Researching COVID to enhance recovery (RECOVER) pregnancy study protocol: Rationale, objectives, and designPLOS ONE

Dear Dr. Metz,

Thank you for submitting your manuscript to PLOS ONE. After careful consideration, we feel that it has merit but does not fully meet PLOS ONE’s publication criteria as it currently stands. Therefore, we invite you to submit a revised version of the manuscript that addresses the points raised during the review process.

We look forward to receiving your revised manuscript.

Kind regards,

Luis Felipe Reyes, M.D., Ph.D., MSc.

Academic Editor

PLOS ONE

Journal Requirements:

"I have read the journal's policy and the authors of this manuscript have the following competing interests:

Dr. Metz reports personal fees from Pfizer for her role as a medical consultant for a SARS-CoV-2 vaccination in pregnancy study, grants from Pfizer for role as a site PI for SARS-CoV-2 vaccination in pregnancy study, grants from Pfizer for role as a site PI for RSV vaccination in pregnancy study, and grants from Gestvision for role as a site PI for a preeclampsia study outside the submitted work. Dr. Horwitz reported serving as a member of the National Academy of Medicine Committee on the Long-Term Health Effects Stemming from COVID-19 and Implications for the Social Security Administration. Dr. Costantine reported receiving grant support for work not related to this paper from Baxter International and Siemens Healthcare and personal consulting fees not related to this paper from Progenity and Siemens Healthcare."

We note that you received funding from a commercial source: Pfizer, Gestvision, Baxter International, Progenity and Siemens Healthcare

Within this Competing Interests Statement, please confirm that this does not alter your adherence to all PLOS ONE policies on sharing data and materials by including the following statement: "This does not alter our adherence to PLOS ONE policies on sharing data and materials.” (as detailed online in our guide for authors http://journals.plos.org/plosone/s/competing-interests).  If there are restrictions on sharing of data and/or materials, please state these. 

Please note that we cannot proceed with consideration of your article until this information has been declared. 

5. One of the noted authors is a group or consortium: The RECOVER Initiative

In addition to naming the author group, please list the individual authors and affiliations within this group in the acknowledgments section of your manuscript. Please also indicate clearly a lead author for this group along with a contact email address.

8. We note that the original protocol that you have uploaded as a Supporting Information file contains an institutional logo. As this logo is likely copyrighted, we ask that you please remove it from this file and upload an updated version upon resubmission.

9. We note that the original protocol file you uploaded contains a confidentiality notice indicating that the protocol may not be shared publicly or be published. Please note, however, that the PLOS Editorial Policy requires that the original protocol be published alongside your manuscript in the event of acceptance. Please note that should your paper be accepted, all content including the protocol will be published under the Creative Commons Attribution (CC BY) 4.0 license, which means that it will be freely available online, and any third party is permitted to access, download, copy, distribute, and use these materials in any way, even commercially, with proper attribution.

Therefore, we ask that you please seek permission from the study sponsor or body imposing the restriction on sharing this document to publish this protocol under CC BY 4.0 if your work is accepted. We kindly ask that you upload a formal statement signed by an institutional representative clarifying whether you will be able to comply with this policy. Additionally, please upload a clean copy of the protocol with the confidentiality notice (and any copyrighted institutional logos or signatures) removed.

Reviewers' comments:

Reviewer's Responses to Questions

**Comments to the Author**

1. Does the manuscript provide a valid rationale for the proposed study, with clearly identified and justified research questions?

Reviewer #1: Yes

Reviewer #2: Yes

Reviewer #3: Yes

2. Is the protocol technically sound and planned in a manner that will lead to a meaningful outcome and allow testing the stated hypotheses?

Reviewer #1: Yes

Reviewer #2: Yes

Reviewer #3: Yes

3. Is the methodology feasible and described in sufficient detail to allow the work to be replicable?

Reviewer #1: Yes

Reviewer #2: Yes

Reviewer #3: Yes

4. Have the authors described where all data underlying the findings will be made available when the study is complete?

Reviewer #1: No

Reviewer #2: Yes

Reviewer #3: Yes

5. Is the manuscript presented in an intelligible fashion and written in standard English?

Reviewer #1: Yes

Reviewer #2: Yes

Reviewer #3: Yes

6. Review Comments to the Author

You may also provide optional suggestions and comments to authors that they might find helpful in planning their study.

Reviewer #1: In this study protocol, RECOVER-Pregnancy will investigate whether having SARS-CoV-2 during pregnancy modifies the risk factors, prevalence, and phenotype of post-cute sequelae of SARS-CoV-2. Both prospective and retrospective cohorts will be enrolled. Adults with pregnancy during the COVID-19 pandemic and their offspring exposed and unexposed in utero will be included. The expected total sample size is 2,300 (1867 infected / 433 uninfected).

Minor revision:

Identify the software that will be used for the statistical analysis.

Reviewer #2: The authors propose a large scale, multicentre cohort follow up study of pregnant individuals (and their offspring) exposed to Sars-CoV-2 during pregnancy. This proposal is closely related to the RECOVER-Adult and Paediatric cohorts, utilising the same methodological approaches with important modifications to account for pregnancy.

This is an important study, which is appropriately designed. The tiered approach allows for the opportunity to obtain more detailed information and samples from a much smaller cohort, thereby helping to minimise participant burden.

The authors appropriately address the limitations of the design (eg limited ability to gain early assessments from those exposed to the earlier COVID variants) and justify the pregnancy specific modifications.

The team of researchers have significant clinical and research experience to undertake this work.

All key components of the protocol appear to be included in this manuscript. Appropriate management and quality assurance structures appear to be in place. IRB approvals are in place.

Reviewer #3: Thank you for the opportunity to review this submission. The manuscript is well written and the protocol is describe in sufficient detail. I have a few comments for the authors consideration:

1) Line 279: please clarify if men (male sex assigned at birth) are eligible as part of the non-pregnant adult participants . Similarly, whether women who are outside of "traditional" child bearing ages are eligible. With a significantly smaller unexposed compared to exposed group this could raise concerns about how exchangable your exposed and unexposed study samples really are. Based on the sample size presented just prior to this, it may just be the language that needs to be fixed for clarity (i.e., assuming 25% of non-pregnant adult participants are female..).

2) following the above point, it looks like in the sample size calculations you are including unexposed from the recover-adult cohort for a pregnant to non-pregnant comparison and it's still a little confusing as to whether this will be restricted by sex at birth to females only.

3) Statistical analysis section - it looks like you powered everything for a risk difference but most of the stats section related to odds ratios and logistic regression. Could be useful to describe those methods in more detail, right now it reads more as an added rather than primary analysis.

Minor comments:

1) Recommend using the term unexposed instead of controls as you are neither randomizing nor performing a case-control study.

2) Could be useful to discuss briefly how you will handle vaccination status in this.

7. PLOS authors have the option to publish the peer review history of their article (what does this mean?). If published, this will include your full peer review and any attached files.

Reviewer #1: No

Reviewer #2: No

Reviewer #3: No

---

## [Author Response · Author response to Decision Letter 0]

9 Oct 2023

Reviewer #1: In this study protocol, RECOVER-Pregnancy will investigate whether having SARS-CoV-2 during pregnancy modifies the risk factors, prevalence, and phenotype of post-cute sequelae of SARS-CoV-2. Both prospective and retrospective cohorts will be enrolled. Adults with pregnancy during the COVID-19 pandemic and their offspring exposed and unexposed in utero will be included. The expected total sample size is 2,300 (1867 infected / 433 uninfected).

Minor revision:

Identify the software that will be used for the statistical analysis.

Response: R software will be used for statistical analyses. This information has been added to the manuscript.

“Statistical analyses will be performed using R software.”

Reviewer #2: The authors propose a large scale, multicentre cohort follow up study of pregnant individuals (and their offspring) exposed to Sars-CoV-2 during pregnancy. This proposal is closely related to the RECOVER-Adult and Paediatric cohorts, utilising the same methodological approaches with important modifications to account for pregnancy.

This is an important study, which is appropriately designed. The tiered approach allows for the opportunity to obtain more detailed information and samples from a much smaller cohort, thereby helping to minimise participant burden.

The authors appropriately address the limitations of the design (eg limited ability to gain early assessments from those exposed to the earlier COVID variants) and justify the pregnancy specific modifications.

The team of researchers have significant clinical and research experience to undertake this work.

All key components of the protocol appear to be included in this manuscript. Appropriate management and quality assurance structures appear to be in place. IRB approvals are in place.

Response: Thank you. No revisions needed in response to this review.

Reviewer #3: Thank you for the opportunity to review this submission. The manuscript is well written and the protocol is describe in sufficient detail. I have a few comments for the authors consideration:

1) Line 279: please clarify if men (male sex assigned at birth) are eligible as part of the non-pregnant adult participants . Similarly, whether women who are outside of "traditional" child bearing ages are eligible. With a significantly smaller unexposed compared to exposed group this could raise concerns about how exchangable your exposed and unexposed study samples really are. Based on the sample size presented just prior to this, it may just be the language that needs to be fixed for clarity (i.e., assuming 25% of non-pregnant adult participants are female..).

Response: We have clarified that comparisons are planned with non-pregnant female participants in the RECOVER-Adult Cohort on p. 8, 9, 10 and 26.

Thus, it remains unknown if there is a differential likelihood of PASC among individuals who contract SARS-CoV-2 during pregnancy compared with non-pregnant reproductive age females. 

We hypothesize that the incidence of persistent long-term sequelae among pregnant people will vary from that of non-pregnant reproductive age females given the unique changes in physiology, hormone levels, and inflammatory milieu in pregnancy.

…and (3) evaluating if contracting SARS-CoV-2 in pregnancy is associated with a differential risk of developing PASC compared with non-pregnant reproductive age females.

Differences in the prevalence or symptoms of PASC among those who acquire SARS-CoV-2 in pregnancy compared with non-pregnant reproductive age females could help with determination of the underlying pathophysiology of PASC. 

2) following the above point, it looks like in the sample size calculations you are including unexposed from the recover-adult cohort for a pregnant to non-pregnant comparison and it's still a little confusing as to whether this will be restricted by sex at birth to females only.

Response: Text was further modified for clarity on p. 13. There is also a statement on p.24 clarifying that comparisons will be made to non-pregnant reproductive age females.

Assuming 25% of non-pregnant adult participants are female and of reproductive age, the minimum detectable PASC risk difference between SARS-CoV-2 infected reproductive age female participants with and without active pregnancy at infection is 4.7% (8.9% in a 25% subgroup).

Logistic regression analyses will be conducted to investigate associations between cumulative incidence of PASC and clinical factors including vaccination status, demographics, and social determinants of health among participants with infection during pregnancy, as well as comparing those infected during pregnancy with infected non-pregnant reproductive age females.

3) Statistical analysis section - it looks like you powered everything for a risk difference but most of the stats section related to odds ratios and logistic regression. Could be useful to describe those methods in more detail, right now it reads more as an added rather than primary analysis.

Response: The sample size and power calculations on pp. 13-15 report minimum detectable differences in PASC risk due to their ease of interpretation, though as the reviewer notes our primary analyses of PASC risk are planned as logistic regression models which estimate odds ratios. This choice does not affect the design of the study, as minimum detectable risk differences can be equivalently expressed as minimum detectable odds ratios to correspond with the planned logistic regression analyses. For completeness, we have updated Table 1 to report both sets of minimum detectable effects.

Minor comments:

1) Recommend using the term unexposed instead of controls as you are neither randomizing nor performing a case-control study.

Response: Thank you for this suggestion. All instances of “control” were changed to “unexposed” or “uninfected participants”.

2) Could be useful to discuss briefly how you will handle vaccination status in this.

Response: Vaccination status will be considered when appropriate in analyses related to the development of PASC. This covariate for modeling is now specifically mentioned on p. 24.

Logistic regression analyses will be conducted to investigate associations between cumulative incidence of PASC and clinical factors including vaccination status, demographics, and social determinants of health among participants with infection during pregnancy, as well as comparing those infected during pregnancy with infected non-pregnant reproductive age females.

---

## [Editor Report · Decision Letter 1]

12 Oct 2023

Researching COVID to enhance recovery (RECOVER) pregnancy study: Rationale, objectives, and design

PONE-D-23-11234R1

Dear Dr. Metz,

We’re pleased to inform you that your manuscript has been judged scientifically suitable for publication and will be formally accepted for publication once it meets all outstanding technical requirements.

Kind regards,

Luis Felipe Reyes, M.D., Ph.D., MSc.

Academic Editor

PLOS ONE
---

## [Editor Report · Acceptance letter]

11 Dec 2023

PONE-D-23-11234R1 

Researching COVID to enhance recovery (RECOVER) pregnancy study: Rationale, objectives and design 

Dear Dr. Metz:

I'm pleased to inform you that your manuscript has been deemed suitable for publication in PLOS ONE. Congratulations! Your manuscript is now with our production department. 

Kind regards, 

on behalf of

Dr. Luis Felipe Reyes 

Academic Editor

PLOS ONE